# Steady motion of 80-nm-size skyrmions in a 100-nm-wide track

Dongsheng Song [1,2] ✉, Weiwei Wang [1,2], Shuisen Zhang [2,3], Yizhou Liu[2], Ning Wang[2], Fengshan Zheng [4,5], Mingliang Tian [2,6,7], Rafal E. Dunin-Borkowski [4], Jiadong Zang [8,9] & Haifeng Du [1,2,6] ✉

The current-driven movement of magnetic skyrmions along a nanostripe is essential for the advancement and functionality of a new category of spin-tronic devices resembling racetracks. Despite extensive research into sky-rmion dynamics, experimental verification of current-induced motion of ultra-small skyrmions within an ultrathin nanostripe is still pending. Here, we unveil the motion of individual 80 nm-size skyrmions in an FeGe track with an ultrathin width of 100 nm. The skyrmions can move steadily along the track over a broad range of current densities by using controlled pulse durations of as low as 2 ns. The potential landscape, arising from the magnetic edge twists in such a geometrically confined system, introduces skyrmion inertia and ensures efficient motion with a vanishing skyrmion Hall angle. Our results showcase the steady motion of skyrmions in an ultrathin track, offering a practical pathway for implementing skyrmion-based spintronic devices.

Magnetic skyrmions are particle-like spin textures with non-trivial topology characterized by the integer valued topological charge $Q = \frac{1}{4\pi} \int \boldsymbol{n} \cdot (\partial_x \boldsymbol{n} \times \partial_y \boldsymbol{n}) dx dy$, where $\boldsymbol{n}(x,y)$ is the unit vector along the spin direction[1–3]. The advantageous features of skyrmions, such as their nanoscale sizes ranging from 3–150 nm and high topological stability[3,4], along with the ultralow current density required for the movement of skyrmion lattices by spin transfer torques (STT)[5,6], make them pro-mising candidates for utilization in racetrack-like spintronic devices[7]. This has led to numerous studies exploring applications in the emer-ging field of skyrmionics[6,8–10]. Within this realm, the stable and con-trollable motion of skyrmions along a track serves as the cornerstone for the practical implementation of potential skyrmion-based devices.

However, two crucial challenges remain unresolved. The first challenge is to achieve controlled motion of nanometer-sized skyrmions in ultrathin nanotracks, meeting the high-density requirements of modern microelectronic devices. Although numer-ical investigations into the current-induced dynamics of highly geo-metrically confined skyrmions in sub-100 nm nanostripes has been conducted[11,12], experimental progress has plateaued over the past decade due to the difficulty of both fabricating ultrathin nanotracks and performing in situ observations of the motion of small skyrmions within them. To date, only isolated skyrmions with diameters of a few hundred nanometers or small skyrmions moving at an angle have been realized in nanostripes[13–18]. The second challenge involves eliminating the skyrmion Hall effect[19–21] to prevent skyrmion annihi-lation at the track edges due to the unit-$Q$-induced Magnus force. Despite significant efforts to suppress the skyrmion Hall angle (SHA), such as employing the synthetic antiferromagnetic skyrmions[22,23],

[1]Institutes of Physical Science and Information Technology, Anhui University, Hefei 230601, China. [2]Anhui Province Key Laboratory of Low-Energy Quantum Materials and Devices, High Magnetic Field Laboratory, HFIPS, Chinese Academy of Sciences, Hefei, Anhui 230031, China. [3]University of Science and Technology of China, Hefei 230026, China. [4]Ernst Ruska-Centre for Microscopy and Spectroscopy with Electrons and Peter Grünberg Institute, For-schungszentrum Jülich, 52425 Jülich, Germany. [5]Spin-X Institute, Center for Electron Microscopy, School of Physics and Optoelectronics State Key Laboratory of Luminescent Materials and Devices Guangdong-Hong Kong-Macao Joint Laboratory of Optoelectronic and Magnetic Functional Materials, South China University of Technology, Guangzhou 511442, P. R. China. [6]Science Island Branch of Graduate School, University of Science and Technology of China, Hefei, Anhui 230026, China. [7]School of Physics and Optoelectronic Engineering, Anhui University, Hefei 230601, China. [8]Department of Physics and Astronomy, University of New Hampshire, Durham, NH 03824, USA. [9]Materials Science Program, University of New Hampshire, Durham, NH 03824, USA. ✉e-mail: dsong@ahu.edu.cn; duhf@hmfl.ac.cn

 1

temperature-compensated ferrimagnetic skyrmions[24–26], antiferromagnetic skyrmions[27,28] and skyrmioniums[22], achieving a completely vanished SHA has not been realized experimentally until now.

Here, we demonstrated the capability to engineer the chiral magnet FeGe into a nanostripe with an ultrathin width of $w \sim 100$ nm. Along this nanostripe, we control and observe the motion of individual skyrmions using in situ electrical contacting and Lorentz transmission electron microscopy (TEM). We show that sub-10 ns current pulses can be used to achieve steady motion of a single skyrmion, a skyrmion pair, and a skyrmion chain along the track, and the skyrmion Hall effect is effectively eliminated in this highly confined system.

## Results and discussion

### Ultrathin nanostripe design and in-situ electrical Lorentz TEM setup

FeGe is a prototypical chiral magnet that has a Curie temperature of $T_c \sim 278$ K. The equilibrium period $\lambda$ of the skyrmion lattice in FeGe is ~80 nm, comparable to the size of a single skyrmion therein[29]. Previous studies of phase diagram for skyrmion in a confined nanostripe have shown that a limited stripe size of $w/\lambda \sim 1.2$ is required to stabilize a single skyrmion[30]. Here, we implemented a FeGe nanostripe designed for integration into the in-situ electrical LTEM setup (Fig. 1a), with a length of ~10 $\mu$m and a width $w$ of ~100 nm, by using an in-house procedure[31,32] (Methods, Supplementary Figs. 1 and 2). The nanostripe was tailored to accommodate a single skyrmion within its confines. The edges of the nanostripe were well maintained, with the thickness of amorphous layers below 3 nm, as revealed using high-resolution TEM imaging (Methods, Supplementary Fig. 3). The ends of the nanostripe were connected to a voltage source using Pt electrodes (Fig. 1b). Electrical current pulses were applied along its long axis with a frequency of 1 Hz. The current-driven skyrmion motion was measured at 95 K. Skyrmions were created by current-pulse-induced Joule heating, whereby a higher current density can nucleate magnetic skyrmions in the presence of a moderate magnetic field (Supplementary Fig. 4). In this way, a variety of skyrmion configurations could be created in the nanostripe, including single skyrmions, skyrmion pairs and skyrmion chains. Here, a single skyrmion is typically imaged as a bright white dot using Lorentz magnetic contrast (Fig. 1c).

### Current-driven motion of a single skyrmion in a 100-nm-wide FeGe nanostripe

A representative trajectory of the motion of an individual skyrmion is shown in Fig. 2a for a pulse duration of $\tau = 5$ ns and a current density of $j = 8.47 \times 10^{10}$ A · m$^{-2}$. When stimulated by ultrashort current pulses, a skyrmion could be driven steadily and straight without being annihilated at the edges of the nanostripe (Supplementary Movie 1). The skyrmion's size and shape were maintained after applying each current pulse. Constrained within the fixed boundaries of the nanostripe, the moving direction aligned collinear with the flow of conduction electrons, demonstrating one-dimensional motion with the absence of the skyrmion Hall effect. Slight deviations in the skyrmion position were resulted from the presence of random pinning sites in the nanostripe. Upon reversing the direction of the current, the skyrmion was subjected to an opposite driving force and returned in a similar manner (Supplementary Movie 2).

Steady motion of individual skyrmions was observed over a wide range of current densities. The $j$-$v$ curve was constructed from the trajectories observed at various current densities, revealing several distinct régimes of skyrmion motion (Fig. 2b). Initially, a single skyrmion was pinned by material defects up to a critical current density of $j_c = 4.24 \times 10^{10}$ A · m$^{-2}$. Slightly above $j_c$, a steep increase in velocity was observed, corresponding to a creep and depinning process caused by intrinsic defects in the material. Skyrmion motion then entered a flow régime, with the velocity increasing linearly over a wide range of current densities to a maximum of 50 m · s$^{-1}$ at almost $j = 8.69 \times 10^{10}$ A · m$^{-2}$. A further increase in current density would cause the instability or annihilation of skyrmion due to unavoidable Joule heating (Supplementary Movie 3).

The dependence of single skyrmion motion on pulse duration was investigated (Fig. 2c). Similarly, creep and depinning, flow régimes are observed at different pulse durations. The skyrmion can be effectively driven even at $\tau = 2$ ns (Supplementary Fig. 5 and Supplementary Movie 4). The highest achieved skyrmion velocity as a function of pulse duration is summarized in the inset to Fig. 2c, reaching a value of 90 m · s$^{-1}$ at $\tau = 2$ ns. These observations highlight the efficient motion of skyrmions in such a highly-confined system. The critical current density $j_c$ decreases exponentially with increasing pulse duration (Fig. 2d). At $\tau = 2$ ns, $j_c$ is ~ $7.89 \times 10^{10}$ A · m$^{-2}$. It is reduced to ~ $3.67 \times 10^{10}$ A · m$^{-2}$ at $\tau = 8$ ns. Below 2 ns, the skyrmion is pinned more easily (Supplementary Fig. 6 and Supplementary Movie 5). Depinning of skyrmions is influenced by both the current density and the pulse width. The phenomenon of skyrmion being more easily pinned for smaller current pulses can indeed be understood through the picture of pinning effects. The pinning centers can be considered as effective potential wells with a characteristic size. Moving the skyrmion's center across this pinning potential takes time, which leads to easier pinning when shorter current pulses are used. The fundamental lower limit on pulse length is contingent on the density and strength of local pinning centers in real material.

The flow régime is extended to high current density with decreasing pulse duration, as the effect of Joule heating effect is

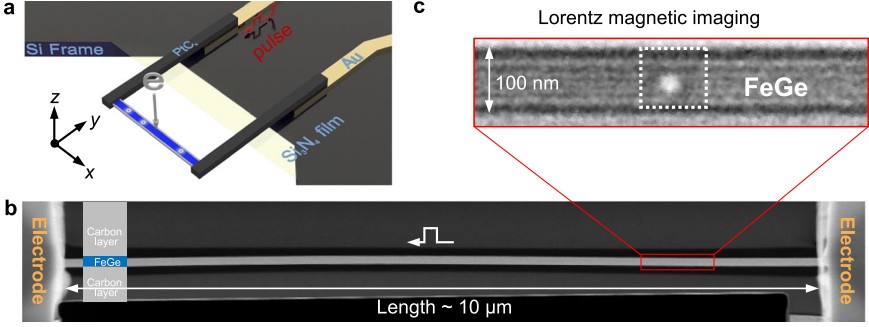

**Fig. 1 | Single skyrmion in a 100-nm-wide FeGe nanostripe. a** Schematic illustration of the FeGe microdevice. The left and right pads represent the Pt electrodes connected to the source of nano-second current pulse through the Au wires in the Si$_3$N$_4$/Si film. The electron beam is along $z$ direction. The single chain skyrmions are hosted in an ultrathin FeGe track. **b** Low magnification high-angle annular dark-field (HAADF) image of an FeGe nanostripe with a length of ~10 $\mu$m and a width of ~100 nm. The schematic illustrates that the upper and lower sides of the FeGe nanostripe are protected by carbon layers. **c** Lorentz TEM image of a single skyrmion in the nanostripe at a defocus value of 600 $\mu$m in the presence of a magnetic field of 234 mT.

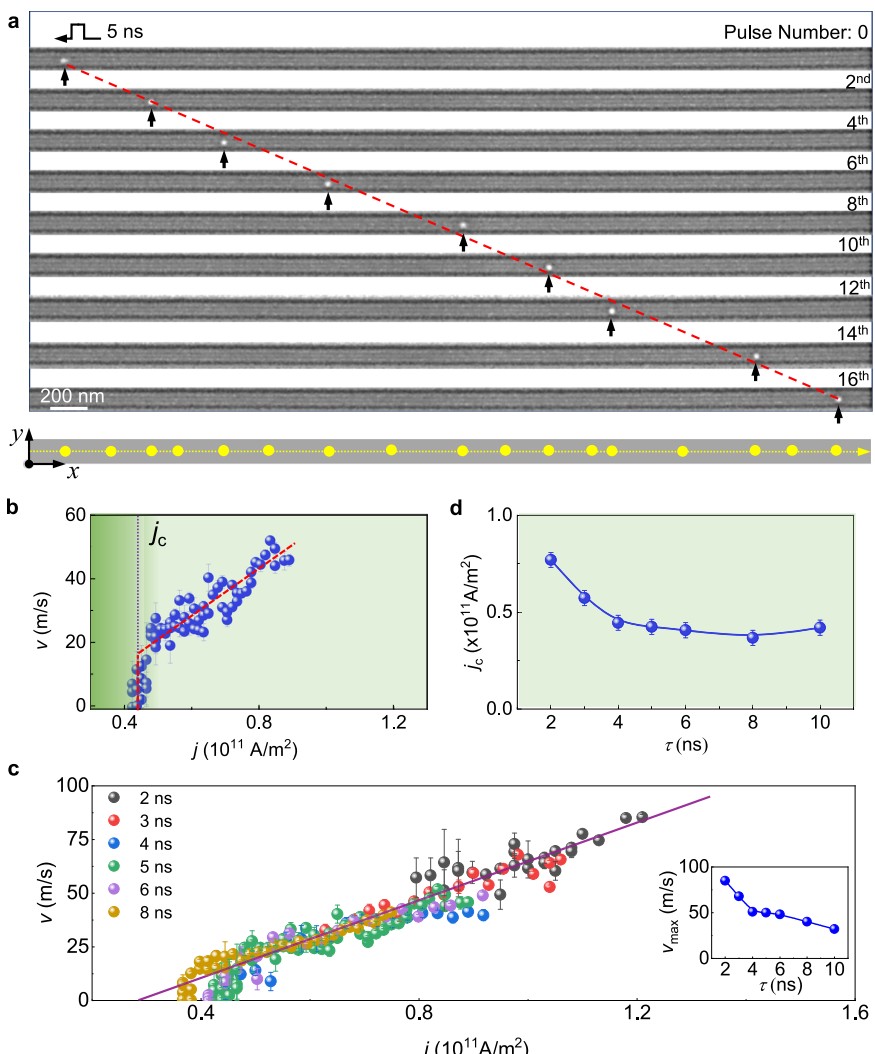

**Fig. 2 | Single skyrmion motion in a 100-nm-wide nanostripe stimulated by successive current pulses. a** Representative snapshots of the current-induced motion of an individual skyrmion. The current density is $j = 8.47 \times 10^{10}$ A · m$^{-2}$ and the pulse duration is $\tau = 5$ ns. The positions of the skyrmion are indicated by black arrows. Trajectories of the skyrmion are summarized by the yellow dots. **b** Skyrmion velocity plotted as a function of current density at $\tau = 5$ ns. The creep and depinning, flow régimes are indicated by dotted red lines. The critical current densities for skyrmion motion ($j_c$) is marked by a vertical dotted black line. **c** $j–v$ relation for a single skyrmion at the pulse durations of 2, 3, 4, 5, 6 and 8 ns. The purple line indicates a fit to the flow régime with a slope of -8.0 ($\times 10^{-10}$ m$^3$/A · s). The inset shows the maximum velocity achieved for different values of $\tau$. **d** Critical current density $j_c$ plotted as a function of pulse duration $\tau$.

strongly reduced. In the flow régime, the same linear relationship with respect to current density is maintained for all pulse durations, as indicated by the purple solid line (Fig. 2c). Within the framework of Thiele's equation for skyrmion dynamics in a confined nanostripe[11,12,33], the slope of the $j–v$ curve that determines the efficiency is governed by the ratio of the nonadiabatic STT coefficient $\beta$ and the Gilbert damping parameter $\alpha$. Through the relationship of $\Delta v_x / \Delta j = - b\beta/\alpha$ (Supplementary Note I), where $b = -\frac{gP\mu_B}{2eM_s}$, $g$ is the Landé factor, $\mu_B$ is the Bohr magneton, $M_s$ is the saturation magnetization, and $P$ is the spin polarization ratio[33], we straightforwardly get $|b\beta/\alpha| \sim 8.0 \, (\times 10^{-10} \, \text{m}^3/\text{A} \cdot \text{s})$, a value considerably larger than the theoretical prediction[11,33]. A large slope means that a skyrmion can be driven with high speed at a low current density in the one-dimensional scenario, offering great advantages for operation with ultra-low energy cost.

We have also investigated the current-driven skyrmion motion under different magnetic fields, as shown in Supplementary Fig. 7. In these cases, skyrmions can still move steadily and straight along the track without experiencing the skyrmion Hall effect. The current-velocity curves exhibit some variations while the linear relationship is still well maintained. At other magnetic fields, the available range of current density is considerably limited in the experiments and a stable current-velocity relationship is not established. However, the critical current density $j_c$ for skyrmion motion is significantly influenced by the magnetic field. As shown in Supplementary Fig. 7a, $j_c$ initially decreases and then increases with respect to the magnetic field. The $j_c$ is mainly influenced by the pinning forces both from the disorders in the sample and the skyrmion-edge interaction. At lower magnetic fields, the pinning force from the edges dominates, and it decreases with the increasing magnetic field due to the attenuated skyrmion-edge interaction, leading to a decrease in $j_c$. By further increasing the magnetic field, the skyrmion-edge interaction weakens, and the skyrmion size diminishes. Consequently, skyrmions become more susceptible to pinning by the disorders, leading to an increase in $j_c$.

## Micromagnetic simulations of skyrmion dynamics in an ultra-thin nanostripe

In order to understand the dynamics of a single skyrmion in an ultra-thin nanostripe, micromagnetic simulations were performed using a

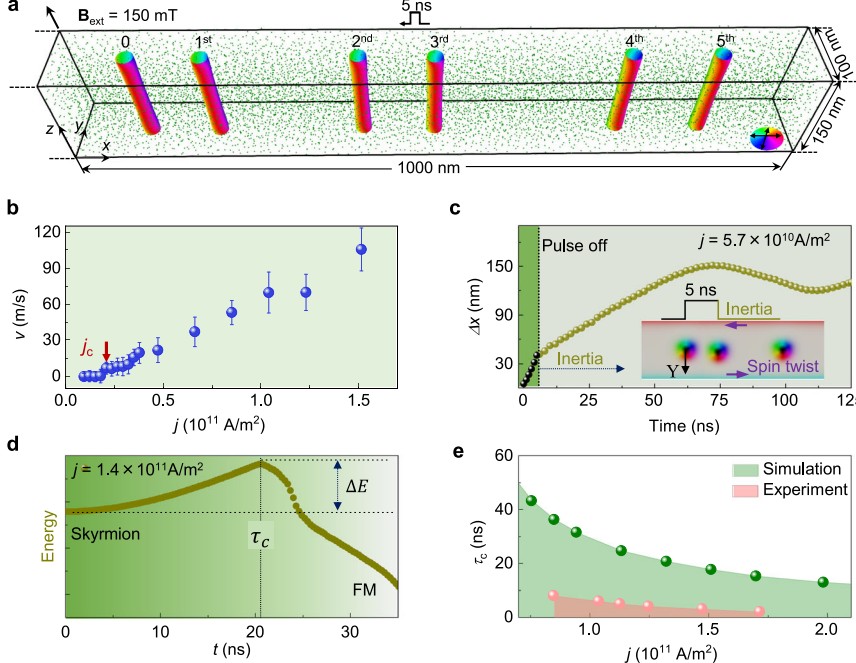

**Fig. 3 | Micromagnetic simulations of skyrmion dynamics in a 100 nm-wide nanostripe. a** Trajectory of a single skyrmion driven by a sequence of current pulses for an external field $H = 150$ mT, a pulse duration $\tau = 5$ ns, a Gilbert damping parameter $\alpha = 0.0167$ and a coefficient $\beta = 0.1336$. The presence of random disorder, which is represented by green dots, is modeled by using an additional anisotropy with $K_u = 2 \times 10^4$ A/m$^3$. The length of the sample is 1000 nm. **b** Calculated skyrmion velocity plotted as a function of current density for a pulse duration of 5 ns. **c** Typical skyrmion displacement $\triangle X$ plotted as a function of time after applying a current pulse of 5 ns for a current density of $j = 5.7 \times 10^{10}$ A/m$^2$. The vertical dashed lines indicate a time of 5 ns. A schematic illustration of skyrmion inertia is shown in the inset. **d** Total system energy plotted as a function of the pulse duration of electrical current. The skyrmion collapses into a ferromagnetic (FM) state above a critical pulse duration. **e** Calculated critical pulse duration $\tau_c$ for skyrmion annihilation at the edge as a function of current density (green points). Experiments were performed in the pink region.

three-dimensional micromagnetic model (Methods). Snapshots of the skyrmion trajectory after applying five pulses at a typical current density of $j = 5.7 \times 10^{10} A \cdot m^{-2}$ are displayed in Fig. 3a and Supplementary Movie 6. The skyrmion exhibits steady motion along the track without apparent skyrmion Hall effect. The observed deviation from equal spacing is attributed to the influence of random pinning effects, as indicated by the presence of green dots. A $j$-$v$ curve is constructed by combining simulated data at different current densities (Fig. 3b). Depinning and flow régimes are observed, similar to the experimental results.

The good agreement between the experimental and theoretical results allows us to obtain further insight into skyrmion dynamics by examining the skyrmion transient trajectory stimulated by each nanosecond pulse. A typical result for $\tau \sim 5$ ns and $j = 5.7 \times 10^{10}$ A/m$^2$ is shown in Fig. 3c. At first, the skyrmion is driven by a horizontal displacement of ~40 nm owing to the current-induced STT effect. A slight vertical displacement (~5 nm) arises because of the skyrmion Hall effect (inset to Fig. 3c). Subsequently, the current pulse is switched off, and the skyrmion begins to return to its equilibrium position, driven by the vertical edge potential imposed by the boundary of nanostripe. This relaxation process leads to skyrmion inertia[34]. The skyrmion then moves forward simultaneously, attaining an additional horizontal displacement of ~85 nm before reaching saturation (Supplementary Movie 7 and Supplementary Fig. 8 for inertia motion in the presence of different current densities).

Phenomenologically, the skyrmion feels an external vertical potential imposed by the magnetic edge twist in confined nanostripe[11,12,35]. In the energy landscape, the potential can be extracted from the total micromagnetic energy of the system and expressed as a harmonic potential $U(Y) = \frac{1}{2}kY^2$, where $Y$ is the $y$-component of the skyrmion's center and $k$ is a constant coefficient (Supplementary Fig. 9). The system energy increases when the

skyrmion deviates from its equilibrium position, and an energy barrier ($\Delta E$) maintains the skyrmion stability (Fig. 3d). The skyrmion would only be annihilated at the edge and collapse into a ferromagnetic (FM) state if a long current pulse is applied to help it cross the barrier ($\Delta E$) (Fig. 3d and Supplementary Movie 8). The characteristic time for annihilating a skyrmion at $j = 1.4 \times 10^{11}$ A/m$^2$ is approximately $\tau_c = 21$ ns (Supplementary Fig. 9 and Supplementary Note II), which is much longer than the pulse duration of 5 ns. Our experiments were conducted in a safe range of current density and pulse duration, as indicated by the pink region (Fig. 3e). Note that the critical current density for skyrmion annihilation due to vertical displacement (green region in Fig. 3e) is much higher than that caused by Joule heating (pink region in Fig. 3e). This indicates that thermal effects play a significant role in setting the upper limit of the current density for this device.

## Single-chain skyrmions and their current-driven motion

At last, the controlled manipulation of various configurations of skyrmion in a nanotrack are demonstrated. In previous theoretical works, the conservation of skyrmion spacing was observed when the initial spacing was very large, and pinning effects were ignored, or skyrmion-skyrmion interactions were neglected[12,35]. However, the steady motion of successive skyrmions is highly dependent on the spacing between them. Almost synchronous motion of successive skyrmions is achieved (Fig. 4a and Supplementary Movie 9) when the two isolated skyrmions are widely spaced apart. As the spacing decreases, prominent skyrmion-skyrmion interactions[36] comes into play. This is expected to result in varying displacements for each skyrmion in the presence of an electrical current, leading to the non-synchronous motion (Supplementary Fig. 10). Therefore, it necessitates a minimum skyrmion-skyrmion spacing to avoid this phenomenon. However, this process is also influenced by pinning effects in real materials, such as the density

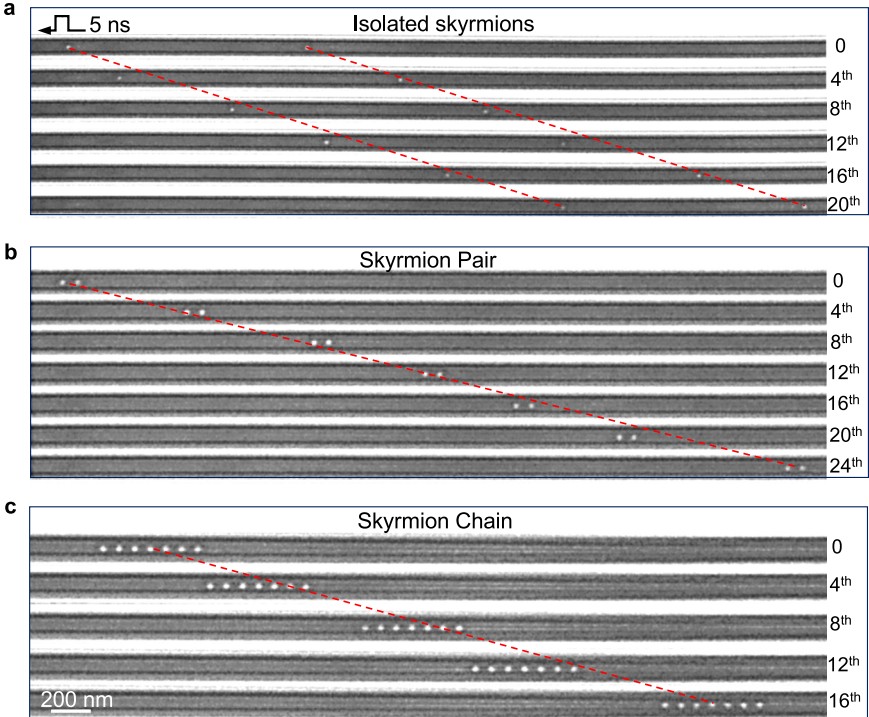

**Fig. 4 | Motion of various configurations of skyrmions in a 100-nm-wide nanostripe.** Snapshots of current-driven motion in (**a**) two isolated skyrmions, (**b**) a skyrmion pair, (**c**) 7 skyrmions using a 5 ns pulse duration and a current density of $j = 7.4 \times 10^{10}$ A $\cdot$ m$^{-2}$. The trajectories are shown using dashed red lines.

and strength of pinning sites, which can likewise induce non-synchronous motion (see Supplementary Note III for the discussion of information-density in the 100 nm-wide nanostripe).

Nevertheless, very-closely-spaced skyrmions could attract one another and form skyrmion pairs or cluster chains[36]. Skyrmion chains featuring various numbers of skyrmions could be steadily driven by currents (Fig. 4b, c, Supplementary Fig. 11, Supplementary Movies 10 and 11), exhibiting the same one-dimensional motion as that observed for a single skyrmion. The measured $j$-$v$ curves are almost consistent across different skyrmion numbers (Supplementary Fig. 12), indicating that both single skyrmion and skyrmion chains follow similar current-driven dynamics. This behavior is further reproduced by micromagnetic simulations. The skyrmion chains with different numbers of skyrmions, where the inter-skyrmion interaction is attractive at this distance, show that the current-driven motion remains translational along the track, following the similar inertia motion (Supplementary Fig. 13). Additionally, the corresponding velocities are almost independent of the number of skyrmions (Supplementary Fig. 14). These observations demonstrate that the skyrmion-skyrmion interaction does not significantly alter the dynamics of the skyrmion chains. These skyrmion chains are stable and well-kept during the motion over a wide range of current densities, hosting the potential for promising device implementation.

In summary, our results have demonstrated the controlled motion of single skyrmions in an ultrathin nanostripe with a high and steady velocity, achieved by manipulating the current density and pulse duration. The skyrmion Hall effect, which has always been considered to be detrimental for skyrmion-based devices, was effectively eliminated in this highly confined system. Our findings address long-standing concerns regarding skyrmion dynamics in ultrathin nanostripes, highlighting the feasibility of encoding single skyrmions in nanotracks for skyrmion-based devices, with potential extensions to magnetic multilayers.

## Methods

### In situ electrical biasing and Lorentz TEM experimental setup

Lorentz magnetic imaging was carried out using a Thermo Fisher Talos F200X TEM operated in the Lorentz mode at 200 kV. The objective lens was switched off to provide a magnetic field-free condition. The out-of-plane magnetic field $B$ was pre-calibrated using a Hall effect sensor positioned in a TEM specimen holder. In situ Lorentz TEM was based on a Gatan holder (Model 616.6 cryotransfer holder, Gatan), which was designed for both in situ electrical biasing and cooling experiments, with a temperature range from 95–380 K. Current pulses were provided by a voltage source (AVR-E3-B-PN-AC22, Avtech Electrosystems Ltd.). Current-driven skyrmion dynamics was measured at 95 K.

The experimental setup for in situ electrical biasing and Lorentz TEM is shown in Supplementary Fig. 1. At the end of the holder in Supplementary Fig. 1a, four electrical ports were designed by Gatan to be connected to a voltage source. In order to build a circuit between the ports and the FeGe sample, a customized electrical chip with four Au electrodes was self-designed, as shown in Supplementary Fig. 1b. The Au pads were connected to the ports using Cu wires, which were manually fixed using silver colloidal paste (only two ports were used in present experiments). The electrical TEM FeGe micro-device was fabricated in the customized chip using FIB milling, as such a chip makes the fabrication process compatible with the conventional FIB lift-out method. A magnified image of the FeGe micro-device on the chip, recorded using a secondary electron signal, is shown in Supplementary Fig. 1c. The Pt electrodes are connected to the Au nanowires through the Pt sticks, which were fabricated using the FIB workstation.

### Fabrication of FeGe micro-devices

FeGe micro-devices tailored for in situ Lorentz TEM were fabricated using an focused ion beam (FIB) SEM dual-beam system (Helios NanoLab 600i; FEI) equipped with a GIS and an Omniprobe 200+ micromanipulator. A customized electrical chip with four Au electrodes was self-designed to make the fabrication process compatible with

the conventional FIB lift-out method. The FeGe membrane was first fabricated with a length of 15 μm and a thickness of 1 μm. It was further thinned to a thickness of ~100 nm, which was the desired width of the FeGe nanostripe. Both sides of the membrane were polished using a small beam current, in order to reduce the surface damage as much as possible. In this way, the edge structures of the nanostripe and their magnetic properties could be retained. The membrane was enclosed in deposited carbon layers and fixed to a customized electrical chip. The carbon protection layers were first deposited on both sides of the nanostripe using an electron beam, followed by an ion beam. These layers were much more insulating (>MΩ) compared to the metallic FeGe nanostripe. As a result, the carbon protection layers did not affect the current density. The overall resistance of the device was ~1000 Ω, primarily due to the contact resistance between the Pt electrodes and the FeGe. The left and right sides of the membrane were connected to two Au electrodes through the deposited Pt layers. The membrane was thinned to a thickness of ~150 nm in the viewing direction of the TEM electron beam. Details of the process are shown in Supplementary Fig. 2.

## Experimental design for FeGe nanostripe

The choice of a 100 nm width for the FeGe nanostripe was primarily based on the width-field magnetic phase diagram for skyrmions in confined FeGe nanostripes[30]. Single skyrmions or skyrmion chains were stabilized within this specific nanostripe width range (~100 nm– ~140 nm). Moreover, to demonstrate an ultimate high density of skyrmions in the radial direction of the nanostripe, a 100 nm width for the FeGe nanostripe was chosen in our experiments. The sample thickness in our experiments was not specifically optimized. Generally, the sample was made electron-transparent to ensure clear Lorentz magnetic imaging contrast. Given that the nanostripe width was very thin, the sample could easily deform during the fabrication process if the thickness was further reduced. This made the experiment extremely challenging. Based on our experimental experience, we opted for a thickness of ~150 nm to prepare a high-quality FeGe nanostripe. The magnetic field was optimized to enable a broad range of current densities for steady skyrmion motion. As shown in Supplementary Fig. 7a, skyrmions can be driven across a spectrum of magnetic fields. The lower critical current density corresponds to the threshold at which the skyrmion begins to move, while the upper critical current density is the point at which the skyrmion is annihilated upon application of the current pulse. Consequently, we primarily used a magnetic field of 234 mT in our experiments. The temperature was maintained at 95 K in our experiments, as this was the lowest temperature and most stable condition for our liquid cooling TEM holder.

## Calculation of skyrmion velocities

In our experiments, we always used 15−30 current pulses to move the skyrmion, ensuring the repeatability of our measurements. The displacements of the skyrmions were measured by tracking their trajectories using a Python-based customized code with a graphical user interface (GUI). The skyrmions were driven through the nanostripe, achieving total displacements of 3 - 6 μm. The velocity of the skyrmion was determined by fitting the displacement-time curve, as illustrated in Supplementary Fig. 15. The error bars for skyrmion velocities were calculated based on the fitting errors of the slope. When the skyrmion motion occurred at lower current densities, it was more likely to be pinned after a short distance of movement. In such cases, the skyrmion was driven through several parts of the nanostripe, and multiple velocities were determined.

By calculating the mean velocities and standard deviations under various current densities and pulse durations (τ), we can quote the range of the optimal current density for consistent displacement. The ratio of skyrmion velocities ($v$) and their standard deviations ($\Delta v$) is plotted in Supplementary Fig. 16 under various current densities and

pusle durations. The optimal current density can be identified by a lower ratio, marked by the green box with a ratio below 5%. Specifically, the optimal current density is approximately between $10 \times 10^{10}$ A/m$^2$ and $12 \times 10^{10}$ A/m$^2$ at $\tau = 2$ ns, between $8 \times 10^{10}$ A/m$^2$ and $11 \times 10^{10}$ A/m$^2$ at $\tau = 3$ ns, between $7 \times 10^{10}$ A/m$^2$ and $9 \times 10^{10}$ A/m$^2$ at $\tau = 4$ ns, between $6.5 \times 10^{10}$ A/m$^2$ and $9 \times 10^{10}$ A/m$^2$ at $\tau = 5$ ns, between $5 \times 10^{10}$ A/m$^2$ and $8.5 \times 10^{10}$ A/m$^2$ at $\tau = 6$ ns, and between $4.5 \times 10^{10}$ A/m$^2$ and $7.5 \times 10^{10}$ A/m$^2$ at $\tau = 8$ ns.

## Estimate of local heating effects

The local heating effects was concerned during our experiments. We utilized COMSOL simulations to evaluate these effects. As shown in Supplementary Fig. 17, with a current density of $j = 13 \times 10^{10}$ A/m$^2$ and a pulse duration of 5 ns, the local temperature reaches 260 K, approaching the Curie temperature of FeGe. This current density is comparable to the critical current density of $j = 12.75 \times 10^{10}$ A/m$^2$ in our experiments, which is required to generate the skyrmions through Joule heating in Supplementary Fig. 4. Our $j$-$v$ curve was measured at current densities below $j = 10 \times 10^{10}$ A/m$^2$ at the pulse duration of 5 ns. At this level, the maximum local temperature reaches ~180 K, far below the Curie temperature. At $j = 7 \times 10^{10}$ A/m$^2$, the maximum temperature increase is only ~40 K. Given the linear relationship between current density and velocity below $j = 10 \times 10^{10}$ A/m$^2$, we believe that the local Joule heating effects are not significant within the reasonable range of current densities in Fig. 2. Additionally, at higher current densities, the signature of local heating became apparent. As shown in Movie 3 at a current density of ~11 × 10$^{10}$ A/m$^2$ and pulse duration of 5 ns, the skyrmion was observed to be annihilated during its motion. At an even higher current density of ~12 × 10$^{10}$ A/m$^2$, the skyrmion was annihilated as soon as the current was applied.

## TEM characterization

High-angle annular dark-field (HAADF) STEM, high-resolution TEM, energy-dispersive X-ray (EDX) and electron energy-loss spectroscopy (EELS) were carried out at 300 kV using a Thermo Fisher Themis Z microscope equipped with a field-emission electron gun, a DCOR probe corrector, a DCOR image corrector, a super-X EDX detector and a Gatan Continuum EELS system. EDX maps were recorded and analyzed using Velox software. High-resolution TEM images were recorded to assess the quality of the nanostripe, as shown in Supplementary Fig. 3a–c. The nanostripe had a sharp interface with the carbon protection layer. The amorphous layers on each side of the nanostripe were observed to have a thickness of below 3 nm. Low-loss EELS was used to measure the thickness of the FeGe nanostripe, which has a uniform profile, as shown in Supplementary Fig. 3d, e.

## Micromagnetic simulations

Micromagnetic simulations were performed using the GPU-supported micromagnetic package MicroMagnetic.jl [37]. The studied system is a nanotrack with a width of 100 nm and a thickness of 150 nm. The total micromagnetic energy is

$$E = \int [A(\nabla \mathbf{m})^2 + D\, \mathbf{m} \cdot (\nabla \times \mathbf{m}) + \mu_0 M_s \mathbf{H} \cdot \mathbf{m}]\, dV + \int K\, m_z^2\, dV_I \,,$$

where $A$ is the exchange constant, $(\nabla \mathbf{m})^2 = (\nabla m_x)^2 + (\nabla m_y)^2 + (\nabla m_z)^2$, $D$ is the bulk DMI constant, $\mathbf{H}$ is the external field, $K$ is the anisotropy strength and $V_I$ denotes the volume with impurities. In the simulation, the parameters for FeGe were chosen as follows: saturation magnetization $M_s = 3.84 \times 10^5$ A/m, exchange constant $A = 3.25 \times 10^{-12}$ J/m$^2$, DMI constant $D = 5.83 \times 10^{-4}$ J/m and external field $H = 150$ mT. The cell size was $2 \times 2 \times 2$ nm$^3$ and periodic boundary conditions were imposed in the $x$-direction. In the presence of currents, the dynamics of magnetization is governed by the LLG equation with spin transfer torque. In the simulation, Gilbert damping $\alpha = 0.0167$ and $\beta = 0.1336$ were

used. In order to estimate the effective spin velocity $\nu_s$, the local spin polarization $P = 0.7$ was used (Supplementary Note I). Pinning effects induced by disorder were modeled using extra anisotropy[33]. The distribution of disorder was generated using Poisson disc sampling for 3D systems, whereby random pinning sites are distributed as evenly as possible in three dimensions. The strength of the uniaxial anisotropy at the pinning sites was $K = 2 \times 10^5 \, \text{A/m}^3$.

Skyrmion inertia motion was reproduced by the micromagnetic simulations, even with the consideration of pinning effects. At a lower current density of $j = 0.95 \times 10^{10} \, \text{A/m}^2$, the skyrmion was first displaced from its original position and then returned to its equilibrium position after long-term strong oscillations in Supplementary Fig. 8a. This relaxation process is caused by the static potential defined by the edge of the nanostripe. However, owing to the strong pinning effects, the overall skyrmion displacement is apparently zero around the critical current density. At a higher current density of $j = 3.8 \times 10^{10} \, \text{A/m}^2$, the skyrmion is effectively driven with a large displacement of ~70 nm in Supplementary Fig. 8b. In this case, the skyrmion is still moving even after switching off the current pulse at 5 ns, which can be understood by the skyrmion inertia.

## Data availability
The data supporting the findings of this study are available within the article and its Supplementary Information files from the corresponding author upon request.

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

## Acknowledgements

This work was supported by the National Key R&D Program of China (Grant No. 2022YFA1403603 (H.D. and D.S.)), the National Natural Science Funds for Distinguished Young Scholar (52325105 (H.D.)), the National Natural Science Foundation of China (12241406 (H.D.), 52173215 (D.S.), 12374098 (W.W.)), the CAS Project for Young Scientists in Basic Research (YSBR-084 (H.D.)), the Strategic Priority Research Program of the Chinese Academy of Sciences (Grant No. XDB33030100 (H.D.)), the Chinese Academy of Sciences under contract No. JZHKYPT-2021-08 (H.D.), the National Natural Science Fund for Excellent Young Scientists Fund Program (Overseas, D.S.), and the Natural Science Foundation of

Anhui Province for Excellent Young Scientist (2108085Y03 (D.S.)). J. Z. were supported by the Office of Basic Energy Sciences, Division of Materials Sciences and Engineering, U.S. Department of Energy, under Award No. DE-SC0020221 and Alexander von Humboldt Foundation (J.Z.); F.Z. and R.E.D-B. acknowledged funding from the European Research Council (ERC) under the European Union's Horizon 2020 research and innovation program (Grant No. 856538, project "3D MAGiC" (F.Z. and R.E.D-B.)).

## Author contributions

H.D. supervised the project. D.S. and H.D. conducted the TEM experiments and data analysis with the contributions from F.Z and R.E.D-B. W.W., S.Z, Y.L. and J.Z. conducted the micromagnetic simulations. H.D., D.S., W.W. and S.Z prepared the manuscript with the input from N.W and M.T.

## Competing interests

The authors declare no competing interests.
