## [Peer Review File · Nature Communications]

Steady motion of 80-nm-size skyrmions in a 100-nm-wide trackREVIEWER COMMENTS

Reviewer #1 (Remarks to the Author):

The authors have provided a direct observation of skyrmion motions in an ultrathin FeGe stripe, employing manipulated current density and pulse duration within a TEM. Through a combination of experimental observations and simulations, they have investigated the current-driven movement of single skyrmions, skyrmion pairs, and chains in detail. Given the novelty and significance of the topic concerning the dynamics of skyrmions, which remains relatively unexplored, the paper is likely to garner wide interest from the magnetic materials research community. Overall, the paper is well-written and can be accepted for publication, with minor revisions.

1. Will the crystallographic orientation affect the movement of skyrmion? Can author give crystal orientation of the FeGe strip along the moving direction?
2. Was carbon protection layer removed after ion milling? Will carbon affect the current density during pulse experiment?

Reviewer #2 (Remarks to the Author):

The results from the 'Steady motion of 80-nm-size skyrmion in a 100-nm-wide track' are truly impressive, especially from a specimen preparation standpoint. For this work, I believe the novelty and difficulty from this work is getting this experiment to work and making the width of the FeGe film thin enough to be 100nm. Furthermore, this result will aid future work so that encoding single skyrmions in nanotracks becomes more feasible. In addition, I have some scientific questions regarding the paper.

- 1) This work shows that a skyrmion whose size is 80 nm can exist at $T \sim 278\text{K}$. For FeGe, there could be a range of differing magnetic phases that are dependent on sample thickness and temperature. In particular, why was 100 nm film width and corresponding 80 nm skyrmion size chosen over a variety of $w/\lambda \sim 1.2$ ratios. Was there a prediction that it gave the best results?
- 2) How is the dynamics of the skyrmion related to its size? Was this chosen to match a particular current density and the corresponding pulse duration?
- 3) Furthermore, in the case of multiple skyrmions (such as skyrmion pair and skyrmion chain in Fig 4), are there interactions between the adjacent skyrmions that could also be modeled? Do these interactions slow down the skyrmion motion? Is there a model that describes interactions between adjacent skyrmions phenomenologically as they are moving? For example, the motion of two masses with a spring attached moving together translationally and rotationally.
- 4) Skyrmions because of their vorticity follow rotational dynamics where there is translation and rotational motion. From the paper, it seems like the current-driven is only given for the translational motion. Is it possible to model the rotational motion of the skyrmions?
- 5) In your system, please explain how you were able to generate and isolate 1 skyrmion, 2 skyrmions and a chain of skyrmions. Is it through increasing the applied magnetic field or changes in temperatures?
- 6) It seems like the resulting moment of inertia of the skyrmion, depending on the applied current density, allows the skyrmion to either stay on the track or get annihilated at the edges as it moves more in y than x. In that case, a wider racetrack maybe more feasible for its device application. Is this true or will a wider racetrack generate larger skyrmions if the

thickness of the film is kept the same?

Reviewer #3 (Remarks to the Author):

Using real-space magnetic imaging, the authors demonstrate efficient current-driven motion of single skyrmions and skyrmion chains in a confined FeGe nanotrack, subsequently eliminating the skyrmion Hall effect previously considered to be a major problem for device applications. They provide a thorough and straightforward characterisation of the one-dimensional skyrmion motion and demonstrate skyrmion velocities up to 90 ms⁻¹. These results are then backed up with micromagnetic simulations which discuss the presence of a skyrmion inertia effect and the limiting parameters for skyrmion motion without annihilation. Finally, the authors extend the discussion into motion of multiple skyrmions closely bound together.

In general, the manuscript provides a new demonstration of one-dimensional skyrmion motion on a scale suitable for device applications and seems suitable for publication within Nature Communications. However, before this paper can be recommended for publication there are several points the authors may wish to clarify:

1. Could the authors comment on the repeatability of their measurements? This is important with regard to error analysis.
2. Was there any signature of the local heating caused by the current pulses and subsequent skyrmion motion in Fig 2? Has any attempt been made to estimate the local heating?
3. What is the explanation for the skyrmion being pinned more easily for smaller current pulses (Fig. S6)? Is there a fundamental lower limit on pulse-length required to drive the skyrmion motion?
4. The ratio of nonadiabatic spin-transfer torque and Gilbert damping parameter is noted to be considerably larger than the theoretical prediction. Can the authors explain the significance of this and why the experimental determination is so significantly larger than previous theory?
5. Does the current-velocity relation change for the cases of skyrmion chains and how does this change with an increasing number of skyrmions?
6. How does the skyrmion inertia change for a collective skyrmion chain? Can the inertia still be considered for the single skyrmion or does the object need to be considered collectively?
7. For the collective motion of a skyrmion chain (Fig. S9), the skyrmion-skyrmion interactions leads to non-synchronous motion which would likely lead to difficulties in extending this to racetrack-type device schemes. Can the authors determine a minimum skyrmion-skyrmion spacing before this non-synchronous motion occurs and therefore state a maximum information-density possible within this nanotrack regime?
8. In 496080_0_video_73316_s9wnwz.mp4 and 496080_0_video_73317_s9wnx0.mp4, at the 14th and 2nd pulses respectively, it looks like the skyrmion splits into two. Can the authors explain this?

RE: Manuscript Number: NCOMMS-24-17355-T

“Steady motion of 80-nm-size skyrmions in a 100-nm-wide track” by Dongsheng Song et al.

Response to reviewers

We would like to express our gratitude to the reviewers for their diligent examination and insightful feedback on our work. The revised manuscript has been crafted, incorporating all the comments and suggestions provided by the reviewers. We look forward to continued support from the reviewers.

For your convenience, detailed point-to-point responses to all raised comments are presented below. All page numbers, references, and figures correspond to the revised version of the manuscript. Changes have been indicated in red throughout the revised manuscript.

Sincerely yours,

Haifeng Du (Prof.) on behalf of all of the authors

High Magnetic Field Laboratory of Chinese Academy of Sciences

Hefei Institutes of Physical Sciences

No. 350 Shushanhu Road, Hefei, 230031, Anhui Province, China

E-mail: duhf@hmfl.ac.cn

Reviewer #1 (Remarks to the Author):

The authors have provided a direct observation of skyrmion motions in an ultrathin FeGe stripe, employing manipulated current density and pulse duration within a TEM. Through a combination of experimental observations and simulations, they have investigated the current-driven movement of single skyrmions, skyrmion pairs, and chains in detail. Given the novelty and significance of the topic concerning the dynamics of skyrmions, which remains relatively unexplored, the paper is likely to garner wide interest from the magnetic materials research community. Overall, the paper is well-written and can be accepted for publication, with minor revisions.

Response:

We thank the reviewer for his/her judgement of the merit of our work and positive comments. The revisions have been made point by point in the following to further improve the manuscript.

Comment 1: Will the crystallographic orientation affect the movement of skyrmion? Can author give crystal orientation of the FeGe strip along the moving direction?

Response:

We thank the reviewer for his/her comment and suggestion. The crystallographic orientation does not affect the movement of skyrmions because the chiral magnet FeGe has very weak magnetic anisotropy. Indeed, we have fabricated many nanostripes in our experiments, given the challenging nature of these experiments. We are not intended to control the sample orientation, and we did not observe obvious differences in skyrmion motion across these nanostripes.

Additionally, as suggested by the reviewer, we have indicated the crystal orientation based on the high-resolution TEM images, as shown in **Fig. R1** below. The orientation of the nanostripe along the electron beam is close to the $[210]$ direction, and the direction of skyrmion movement is close to the $[1-22]$ direction.

Fig. R1 | Characterization of the FeGe nanostripe sample. **a**, Low magnification TEM image of the FeGe nanostripe. **b**, **c**, High-resolution TEM images of the upper and lower interfaces between the FeGe nanostripe and the carbon layers. The amorphous layer has a thickness of ~ 3 nm. **d**, The fast Fourier transformation (FFT) of the TEM image in **c**. The diffraction reflections and crystallographic orientation are indicated in **a**

and **d**, respectively. **e**, Corresponding relative thickness map of the FeGe nanostructure measured using low-loss EELS. **f**, The relative thickness (t) profile is plotted with the horizontal direction along x axis.

Revision:

The information of crystallographic orientation has been incorporated into **Supplementary Fig. 3** in the revised supplementary information.

Comment 2: Was carbon protection layer removed after ion milling? Will carbon affect the current density during pulse experiment?

Response:

We thank the reviewer for his/her comments. We apologize for the oversight in **Fig. 1b**, where the carbon protection layers were not depicted. These carbon layers, initially deposited by electron beam (deep dark region in **Fig. R2b**) and subsequently by ion beam (dark region in **Fig. R2b**) during the FIB process, were not removed after ion milling, as shown in the schematic in **Fig. R2** below.

During sample preparation, the carbon protection layers were first deposited using an electron beam, followed by an ion beam. These layers are much more insulating ($> M\Omega$ in our experiments) compared to the metallic FeGe nanostructure. In our experiments, the overall resistance of the device is approximately 1000Ω , primarily due to the contact resistance between the Pt electrodes and the FeGe. As such, the carbon protection layers do not affect the current density.

Fig. R2 | Single skyrmion in a 100-nm-wide FeGe nanostructure. **a**, Schematic illustration of the FeGe microdevice. The left and right pads represent the Pt electrodes connected to the source of nano-second current pulse through the Au wires in the Si_3N_4/Si film. The electron beam is along z direction. The single chain skyrmions are hosted in an ultrathin FeGe track. **b**, Low magnification high-angle annular dark-field (HAADF) image of an FeGe nanostructure with a length of $\sim 10 \mu m$ and a width of ~ 100 nm. The schematic illustrates that the upper and lower sides of the FeGe nanostructure are protected by carbon layers. **c**, Lorentz TEM image of a single skyrmion in the nanostructure at a defocus value of $600 \mu m$ in the presence of a magnetic field of 234 mT.

Revision:

Fig. 1 has been updated with **Fig. R2** to show the carbon layers in the revised manuscript.

The following statements has been included in the **Method** section, “**Fabrication of FeGe micro-devices**”,

“The carbon protection layers were first deposited on both sides of the nanostructure using an electron beam, followed by an ion beam. These layers were much more insulating ($> M\Omega$) compared to the metallic FeGe nanostructure. As a result, the carbon protection layers did not affect the current density. The overall resistance of the device was approximately 1000Ω , primarily due to the contact resistance between the Pt electrodes and the FeGe.”

Reviewer #2 (Remarks to the Author):

The results from the ‘Steady motion of 80-nm-size skyrmion in a 100-nm-wide track’ are truly impressive, especially from a specimen preparation standpoint. For this work, I believe the novelty and difficulty from this work is getting this experiment to work and making the width of the FeGe film thin enough to be 100nm. Furthermore, this result will aid future work so that encoding single skyrmions in nanotracks becomes more feasible. In addition, I have some scientific questions regarding the paper.

Response:

We thank the reviewer for his/her judgement of the novelty of our work and positive comments which allow us to improve the paper. The revisions have been made point by point in the following to further improve the manuscript.

Comment 1: This work shows that a skyrmion whose size is 80 nm can exist at $T \sim 278$ K. For FeGe, there could be a range of differing magnetic phases that are dependent on sample thickness and temperature. In particular, why was 100 nm film width and corresponding 80 nm skyrmion size chosen over a variety of $w/\lambda \sim 1.2$ ratios. Was there a prediction that it gave the best results?

Response:

We thank the reviewer for his/her comments. The choice of a 100 nm width for the FeGe nanostripe is primarily based on previous work [10.1038/ncomms15569] regarding the width-field magnetic phase diagram for skyrmions in confined FeGe nanostripes. As demonstrated in **Fig. R3a** below, single skyrmions or skyrmion chains are stabilized within a specific nanostripe width range (~ 100 nm - ~ 140 nm). Therefore, to demonstrate an ultimate high density of skyrmions in the radial direction of the nanostripe, a 100 nm width for the FeGe nanostripe was chosen.

Figure 4 | Width-field magnetic phase diagram in a nanostripe. (a) Experimentally observed magnetic states in the wedge-shaped nanostripe in an applied magnetic field and at a fixed temperature $T = 220$ K. The filled squares represent experimental data. Orange, purple, grey and blue squares correspond to helical spiral, single skyrmion chain (SSC), zigzag skyrmion chain (ZSC) and field-saturated states, respectively. Magenta circles correspond to an intermediate metastable state of half skyrmions attached to the edge. Transition lines between domains of corresponding colour are used to guide the eye. (b) Theoretical phase diagram of equilibrium magnetic states calculated for an infinitely long FeGe nanostripe of width W_y and fixed thickness $L = 110$ nm in an applied magnetic field H . The filled circles represent calculated points. The colour notations are the same as in **a**. The phase diagram is extended for an infinite film ($W_y/L \rightarrow \infty$) of corresponding thickness. The upper and right axes are equivalent to the lower and left axes, respectively and are given in reduced units of thickness L and saturation field H_D . The vertical dashed lines in **a,b**, at 30 and 180 nm, mark the limited width of the wedge in the experimental observation. W_y^c and W_y^s correspond to the critical width for ideally circular skyrmions and the transition from a SSC to a ZSC, respectively. The triangular dots are taken from calculations. The dotted line is included as a guide to the eye. Above a certain magnetic field (marked by empty dots) and assuming a fixed skyrmion density, the skyrmions become circular over a range of nanostripe widths, as marked by the dashed region. P_{skl} represents the period of the skyrmion lattice. The dashed region in **b** corresponds to a circular skyrmion.

Fig. R3 | Magnetic phase diagram of width-field in a FeGe nanostripe. [Ref: 10.1038/ncomms15569]

The magnetic field is optimized to enable a broad range of current densities for steady skyrmion motion. As shown in **Fig. R4** below, skyrmions can be driven across a spectrum of magnetic fields. The lower critical current density corresponds to the threshold at which the skyrmion begins to move, while the upper critical current density is the point at which the skyrmion is annihilated upon application of the current pulse. Consequently, we primarily use a magnetic field of 234 mT in our experiments.

Fig. R4 | The lower (motion) and upper (annihilation) critical current density with a pulse duration of 5 ns as a function of magnetic field. The lower critical current density corresponds to the threshold at which the skyrmion begins to move, while the upper critical current density is the point at which the skyrmion is annihilated upon application of the current pulse.

The sample thickness in our experiments is not specifically optimized. Generally, the sample was made electron-transparent to ensure clear Lorentz magnetic imaging contrast. Given that the nanostripe width is very thin, the sample could easily deform during the fabrication process if the thickness is further reduced. This makes the experiment extremely challenging. Based on our experimental experience, we opted for a thickness of approximately 150 nm to prepare a high-quality FeGe nanostripe.

The temperature is maintained at 95 K in our experiments, as this is the lowest temperature and most stable condition for our liquid cooling TEM holder. Our objective here is to demonstrate the possibility of encoding a single skyrmion in an ultrathin nanostripe. We believe that temperature could influence the critical current density, as the pinning effect is temperature-dependent. We appreciate the reviewer’s comments and suggestions. Exploring temperature-dependent dynamics of skyrmions could be the focus of future experiments.

Revision:

The following statements has been involved in the **Method** section (**Experimental design for FeGe nanostripe**) of the revised manuscript to address the experimental design for FeGe nanostripes in our work:

“The choice of a 100 nm width for the FeGe nanostripe was primarily based on the previous work^[30] regarding the width-field magnetic phase diagram for skyrmions in confined FeGe nanostripes. As reported before, single skyrmions or skyrmion chains were stabilized within this specific nanostripe width range (~100 nm ~140 nm). Moreover, to demonstrate an ultimate high density of skyrmions in the radial direction of the nanostripe, a 100 nm width for the FeGe nanostripe was chosen in our experiments. The sample thickness in our experiments was not specifically optimized. Generally, the sample was made electron-transparent to ensure clear Lorentz magnetic imaging contrast. Given that the nanostripe width was very thin, the sample could easily deform during the fabrication process if the thickness was further reduced. This made the experiment extremely challenging. Based on our experimental experience, we opted for a thickness of approximately 150 nm to prepare a high-quality FeGe nanostripe. The magnetic field was optimized to enable a broad range of current densities for steady skyrmion motion. As shown in **Supplementary Fig. 7a**, skyrmions can be driven across a spectrum of magnetic fields. The lower critical current density corresponds to the threshold at which the skyrmion begins to move, while the upper critical current density is the point at which the skyrmion is annihilated upon application of the current pulse. Consequently, we primarily used a magnetic field of 234 mT in our experiments. The temperature was maintained at 95 K in our experiments, as this was the lowest temperature and most stable condition for our liquid cooling TEM holder.”

Comment 2: How is the dynamics of the skyrmion related to its size? Was this chosen to match a particular current density and the corresponding pulse duration?

Response:

We thank the reviewer for his/her comments. In the case of chiral FeGe, the size of the skyrmion is mainly determined by the intrinsic magnetic parameters of the material, usually the ratio between the exchange interaction and the DMI in chiral magnets. The magnetic field can also change the size of skyrmion within a certain range. The size of the skyrmion is not specifically chosen to match a particular current density or pulse duration. As shown in **Fig. 2** of the manuscript, skyrmions can move across a wide range of current densities and pulse durations. However, as described in **Fig. R4** above, we primarily use a magnetic field of 234 mT in our experiments to enable a broad range of current densities for steady skyrmion motion. We have also investigated the current-driven skyrmion motion under several magnetic fields, as shown in **Fig. R5** below. In these cases, skyrmions can still move steadily and straight along the track without experiencing the skyrmion Hall effect. The current-velocity curves exhibit some variations while the linear relationship is still well maintained. At other magnetic fields, the available range of current density is considerably limited (**Fig. R4**) in the experiments and a stable current-velocity relationship is not established.

However, the critical current density j_c for skyrmion motion is significantly influenced by the magnetic field. As shown in **Fig. R4**, j_c initially decreases and then increases with respect to the magnetic field. The j_c is mainly influenced by the pinning forces both from the disorders in the sample and the skyrmion-edge interaction. At lower magnetic fields, the pinning force from the edges dominates, and it decreases with the increasing magnetic field due to the attenuated skyrmion-edge interaction, leading to a decrease in j_c . By further increasing the magnetic field, the skyrmion-edge interaction weakens, and the skyrmion size diminishes. Consequently, skyrmions become more susceptible to pinning by the disorders, leading to an increase in j_c .

Fig. R5 | Skyrmion velocities plotted as a function of current density under different magnetic fields. The pulse duration is 5 ns.

Revision:

The following discussions and related experimental results (**Figs. R4 and R5**) have been involved in the revised manuscript (**Section: Current-driven motion of a single skyrmion in a 100-nm-wide FeGe nanostripe**) as follows,

“We have also investigated the current-driven skyrmion motion under different magnetic fields, as shown in **Supplementary Fig. 7**. In these cases, skyrmions can still move steadily and straight along the track without experiencing the skyrmion Hall effect. The current-velocity curves exhibit some variations while the linear relationship is still well maintained. At other magnetic fields, the available range of current density is considerably limited in the experiments and a stable current-velocity relationship is not established. However, the critical current density j_c for skyrmion motion is significantly influenced by the magnetic field. As shown in **Supplementary Fig. 7a**, j_c initially decreases and then increases with respect to the magnetic field. The j_c is mainly influenced by the pinning forces both from the disorders in the sample and the skyrmion-edge interaction. At lower magnetic fields, the pinning force from the edges dominates, and it decreases with the

increasing magnetic field due to the attenuated skyrmion-edge interaction, leading to a decrease in j_c . By further increasing the magnetic field, the skyrmion-edge interaction weakens, and the skyrmion size diminishes. Consequently, skyrmions become more susceptible to pinning by the disorders, leading to an increase in j_c .”

Comment 3: Furthermore, in the case of multiple skyrmions (such as skyrmion pair and skyrmion chain in Fig 4), are there interactions between the adjacent skyrmions that could also be modeled? Do these interactions slow down the skyrmion motion? Is there a model that describes interactions between adjacent skyrmions phenomenologically as they are moving? For example, the motion of two masses with a spring attached moving together translationally and rotationally.

Response:

We thank the reviewer for his/her insightful comments. The skyrmion-skyrmion interaction has been studied in the same material in a previous work [PRL 120, 197203 (2018)]. It was found that the interaction between skyrmions shows Lennard-Jones-type behavior, i.e., below (above) a critical distance, the skyrmions attract (repel) each other. So far, there has been no systematic study on the dynamics of multiple skyrmions due to the complexity arising from multiple degrees of freedom, which could be very interesting, e.g., they exhibit both translational and rotational dynamics [Nat. Commun. 13, 1593 (2022)] as suggested by the reviewer. Moreover, in a racetrack-type device like the one studied in our work, due to the confinement of the track, the rotational motion is strongly suppressed, and the dynamics of multiple skyrmions reduces to a translational motion along the track.

Furthermore, we have performed additional micromagnetic simulations to investigate whether the skyrmion motion is influenced by the skyrmion-skyrmion interaction. As shown in **Fig. R6** below, for different numbers of skyrmions in the chains, where the inter-skyrmion interaction is attractive at this distance, the current-driven motion of skyrmions remains translational along the track, following similar inertia motion after the current pulse is turned off. The variations in displacements are caused by the randomly distributed pinning sites modeled in the nanostripe. Additionally, the corresponding velocities are almost independent of the number of skyrmions as shown in **Fig. R7**, with variations primarily due to pinning effects and the limited number of current pulses applied in the simulations. Therefore, the skyrmion-skyrmion interaction does not significantly alter the velocity of the skyrmions in this scenario.

Fig. R6 | Skyrmion displacement ΔX plotted as a function of time for skyrmion chains with various skyrmion number (N). The pulse duration is 5 ns and the current density is $j = 10 \times 10^{10}$ A/m². The vertical dashed line indicates a time of 5 ns. The variations in displacements are caused by the randomly distributed pinning sites modeled in the nanostripe.

Fig. R7 | Simulated velocities of skyrmion chains with a various number of skyrmion. The current densities are $5 \times 10^{10} \text{ A/m}^2$ (orange) and $10 \times 10^{10} \text{ A/m}^2$ (green), respectively. The pulse duration is 5 ns. The velocities are averaged over five current pulses in the simulations. The variations are primarily due to pinning effects and the limited number of current pulses applied in the simulations.

Moreover, we have measured the velocities of skyrmion chains with various numbers of skyrmions N ($N = 2, 4$ and 7) in the experiments, and compared them with that of single skyrmions ($N = 1$), as shown in **Fig. R8** below. The j - v curves are almost consistent regardless of the skyrmion number, indicating that both single skyrmion and skyrmion chains take the similar current-driven dynamics.

Fig. R8 | Skyrmion velocities plotted as a function of current density. The skyrmion chains with various skyrmion number are plotted together for comparison. The pulse duration is 5 ns.

Revision:

The following discussions and new results (**Figs. R6, R7 and R8**) have been incorporated in the revised manuscript (**Section: Single-chain skyrmions and their current-driven motion**) and supplementary information as follows,

“Skyrmion chains, featuring various numbers of skyrmions, could be steadily driven by currents (**Fig. 4b-c, Supplementary Fig. 11, Supplementary Videos 10 and 11**), exhibiting the same one-dimensional motion as

that observed for a single skyrmion. The measured j - v curves are almost consistent across different skyrmion numbers (**Supplementary Fig. 12**), indicating that both single skyrmion and skyrmion chains follow similar current-driven dynamics. This behavior is further reproduced by micromagnetic simulations. The skyrmion chains with different numbers of skyrmions, where the inter-skyrmion interaction is attractive at this distance, show that the current-driven motion remains translational along the track, following the similar inertia motion (**Supplementary Fig. 13**). Additionally, the corresponding velocities are almost independent of the number of skyrmions (**Supplementary Fig. 14**). These observations demonstrate that the skyrmion-skyrmion interaction does not significantly alter the dynamics of the skyrmion chains.”

Comment 4: Skyrmions because of their vorticity follow rotational dynamics where there is translation and rotational motion. From the paper, it seems like the current-driven is only given for the translational motion. Is it possible to model the rotational motion of the skyrmions?

Response:

We thank the reviewer for his/her comments. A skyrmion exhibits several dynamical modes, in which the translational and rotational mode are the most straightforward examples, as the reviewer noted. The lowest energy mode of a skyrmion is the translational mode investigated in this work. It is also possible for an electrical current to induce the rotational mode, which is associated with the change in the skyrmion’s helicity. In chiral magnets, the helicity of a skyrmion is fixed by the Dzyaloshinskii-Moriya interaction (DMI) arising from the crystal symmetry and spin-orbit interaction. Any change in the skyrmion’s helicity leads to an increase in the system’s energy. Therefore, the rotational mode of the skyrmion is strongly suppressed due to the energetic considerations in chiral magnets. In centrosymmetric systems, the skyrmion helicity is not fixed due to the absence of DMI. Previous theoretical works, e.g., Nat. Commun. 8, 14394 (2017) and Nat. Commun. 8, 1717 (2017), have already modeled the rotation of skyrmions therein. However, since the chiral magnet FeGe is studied, it is rather difficult to observe the rotational dynamics in the current work.

Comment 5: In your system, please explain how you were able to generate and isolate 1 skyrmion, 2 skyrmions and a chain of skyrmions. Is it through increasing the applied magnetic field or changes in temperatures?

Response:

We thank the reviewer for his/her comments. We apologize for not clearly describing this process in our previous manuscript. The skyrmions were generated through Joule heating effects by applying a higher current under a certain magnetic field. In this scenario, various configurations of skyrmions, including single skyrmions, skyrmion pairs, or skyrmion chains with varying numbers of skyrmions, are randomly generated. These configurations are then isolated by increasing the magnetic field. The detailed procedures for generating and isolating the skyrmions are described as follows:

For single or two skyrmions, they are generated and isolated in the same way as shown in **Fig. R9**. Initially, the sample was saturated with a very high magnetic field, and then the field was reduced to ~ 70 mT to maintain the conical state. Subsequently, an electrical current with a high current density of $j = 12.75 \times 10^{10}$ A/m² and a pulse duration of 5 ns was applied. This process created skyrmions due to Joule heating effects. The current was then turned off and the magnetic field was increased to stabilize the skyrmions. To further reduce the number of skyrmions in the nanostripe, the magnetic field was increased gradually, systematically annihilating skyrmions one by one until only two or a single skyrmion remained. Once this configuration was achieved, the magnetic field was decreased to ~ 234 mT for the current-driven dynamics. This method is straightforward and robust, allowing us to consistently create single skyrmion or two skyrmions in the nanostripe.

Fig. R9 | The typical procedure to generate and isolated two skyrmions in the nanostripe.

While for generating skyrmion chains with varying numbers, the process involves some level of probability. The sample was initialized set to the conical state, similar to the method used for single or two skyrmions above. However, in this case, the magnetic field is maintained at a relatively higher value of ~150 mT. This higher magnetic field is chosen to prevent the creation of too many skyrmions by the thermal effects. By applying a high current density of $j = 12.75 \times 10^{10} A/m^2$, skyrmion chains can be directly generated with a certain probability, as shown in **Fig. R10** with a skyrmion number of 4. However, it should be noted that we cannot control the exact number of skyrmions in the chains in a deterministic manner. This method allows for the creation of varying numbers of skyrmions, but the specific quantity in each chain is subject to variability inherent to the process.

Fig. R10 | The typical procedure to generate a skyrmion chain in the nanostripe.

Revision:

The details for generating and isolating various skyrmion configurations, including **Figs. R9** and **R10**, are involved in the revised supplementary information as follows,

“The skyrmions were generated through Joule heating effects by applying a higher current under a certain magnetic field. The detailed procedures for generating and isolating the skyrmions are described as follows: (1) For single or two skyrmions, they are generated and isolated in the same way as shown in **Supplementary Fig. 4a**. Initially, the sample was saturated with a very high magnetic field, and then the field was reduced to ~ 70 mT to maintain the conical state. Subsequently, an electrical current with a high current density of $j = 12.75 \times 10^{10} A/m^2$ and a pulse duration of 5 ns was applied. This process created skyrmions due to Joule heating effects. The current was then turned off and the magnetic field was increased to stabilize the skyrmions. To further reduce the number of skyrmions in the nanostripe, the magnetic field was increased gradually, systematically annihilating skyrmions one by one until only two or a single skyrmion remained. Once this configuration was achieved, the magnetic field was decreased to ~234 mT for the current-driven dynamics. This method is straightforward and robust, allowing us to consistently create single skyrmion or two skyrmions in the nanostripe. (2) While for generating skyrmion chains with varying numbers, the process involves some level of probability. The sample was initialized set to the conical state, similar to the method used for single or two skyrmions above. However, in this case, the magnetic field is maintained at a relatively higher value of ~150 mT. This higher magnetic field is chosen to prevent the creation of too many skyrmions by the thermal effects. By applying a high current density of $j = 12.75 \times 10^{10} A/m^2$, skyrmion chains can be

directly generated with a certain probability as shown in **Supplementary Fig. 4b** with a skyrmion number of 4. However, it should be noted that we cannot control the exact number of skyrmions in the chains in a deterministic manner. This method allows for the creation of varying numbers of skyrmions, but the specific quantity in each chain is subject to variability inherent to the process.”

Comment 6: It seems like the resulting moment of inertia of the skyrmion, depending on the applied current density, allows the skyrmion to either stay on the track or get annihilated at the edges as it moves more in y than x. In that case, a wider racetrack maybe more feasible for its device application. Is this true or will a wider racetrack generate larger skyrmions if the thickness of the film is kept the same?

Response:

We thank the reviewer for his/her comments. In the case of chiral FeGe, the size of the skyrmion is mainly determined by the intrinsic magnetic parameters of the material (usually the ratio between the exchange interaction and the DMI in chiral magnets) and the geometry of the sample only plays a minor role. So even in a wider racetrack, the skyrmion’s size is approximately the same to that in the sample under investigation (width ~100 nm).

As the reviewer precisely noted, the inertial movement of the skyrmions depends on the applied current density. A larger current density can lead to the annihilation of the skyrmions at the edges. If the racetrack is wider, it may take the skyrmion longer to move towards the edge and acquire the inertia motion. However, the upper limit of the current density is determined by the energy barrier between the skyrmion and ferromagnetic states at the edges. As shown in **Fig. 3e** of the manuscript, the critical current density for skyrmion annihilation due to vertical displacement (green region) is much higher than that caused by Joule heating (pink region). This indicates that thermal effects play a significant role in setting the upper limit of the current density for the device, rather than the width of the nanostripe. Therefore, a wider racetrack would not be more feasible for the device under our investigation.

Fig. 3 | Micromagnetic simulations of skyrmion dynamics in a 100-nm-wide nanostripe. d, Total system energy plotted as a function of the pulse duration of electrical current. The skyrmion collapses into a ferromagnetic (FM) state above a critical pulse duration. **e,** Calculated critical pulse duration τ_c for skyrmion annihilation at the edge as a function of current density (green points). Experiments were performed in the pink region.

Revision:

The following statements have been involved in the revised manuscript (**Section: Micromagnetic simulations of skyrmion dynamics in an ultrathin nanostripe**) to address this point.

“Note that the critical current density for skyrmion annihilation due to vertical displacement (green region in **Fig. 3e**) is much higher than that caused by Joule heating (pink region in **Fig. 3e**). This indicates that thermal effects play a significant role in setting the upper limit of the current density for this device.”

Reviewer #3 (Remarks to the Author):

Using real-space magnetic imaging, the authors demonstrate efficient current-driven motion of single skyrmions and skyrmion chains in a confined FeGe nanotrack, subsequently eliminating the skyrmion Hall effect previously considered to be a major problem for device applications. They provide a thorough and straightforward characterisation of the one-dimensional skyrmion motion and demonstrate skyrmion velocities up to 90 ms⁻¹. These results are then backed up with micromagnetic simulations which discuss the presence of a skyrmion inertia effect and the limiting parameters for skyrmion motion without annihilation. Finally, the authors extend the discussion into motion of multiple skyrmions closely bound together.

In general, the manuscript provides a new demonstration of one-dimensional skyrmion motion on a scale suitable for device applications and seems suitable for publication within Nature Communications. However, before this paper can be recommended for publication there are several points the authors may wish to clarify.

Response:

We thank the reviewer for his/her judgement of the novelty of our work and positive comments which allow us to improve the paper. The revisions have been made point by point in the following to further improve the manuscript.

Comment 1: Could the authors comment on the repeatability of their measurements? This is important with regard to error analysis.

Response:

We thank the reviewer for his/her comment. In our experiments, we used 15-30 current pulses to move the skyrmion, ensuring the repeatability of our measurements, as typically shown in **Fig. 2** and **Videos 1-5**. The displacements of the skyrmions were measured by tracking their trajectories using a Python-based customized code with a graphical user interface (GUI). The skyrmions were driven through the nanostripe, achieving total displacements of 3~6 μm . The velocity of the skyrmion was determined by fitting the displacement-time curve, as illustrated in **Fig. R11** below. The error bars for skyrmion velocities were calculated based on the fitting errors of the slope. When the skyrmion motion occurred at lower current densities, it was more likely to be pinned after a short distance of movement. In such cases, the skyrmion was driven through several parts of the nanostripe, and multiple velocities were determined as depicted in **Fig. 2b** and **2c** at around the critical current density.

Fig. R11 | Skyrmion displacement as a function of pulse time. The lines in different colors correspond to the skyrmion motion under varied current densities. The pulse duration is 5 ns here.

Revision:

The following statements and **Fig. R11** have been incorporated in the **Methods** part and supplementary information.

“Calculation of skyrmion velocities. In our experiments, we always used 15-30 current pulses to move the skyrmion, ensuring the repeatability of our measurements. The displacements of the skyrmions were measured by tracking their trajectories using a Python-based customized code with a graphical user interface (GUI). The skyrmions were driven through the nanostripe, achieving total displacements of 3~6 μm . The velocity of the skyrmion was determined by fitting the displacement-time curve, as illustrated in **Supplementary Fig. 15**. The error bars for skyrmion velocities were calculated based on the fitting errors of the slope. When the skyrmion motion occurred at lower current densities, it was more likely to be pinned after a short distance of movement. In such cases, the skyrmion was driven through several parts of the nanostripe, and multiple velocities were determined.”

Comment 2: Was there any signature of the local heating caused by the current pulses and subsequent skyrmion motion in Fig 2? Has any attempt been made to estimate the local heating?

Response:

We thank the reviewer for his/her comments. Indeed, we were concerned about local heating effects during our experiments. Direct measurement of temperature changes within such a short current pulse is challenging; therefore, we utilized COMSOL simulations to evaluate these effects. As shown in **Fig. R12** below, with a current density of $j = 13 \times 10^{10} \text{ A/m}^2$ and a pulse duration of 5 ns, the local temperature reaches 260 K, approaching the Curie temperature of FeGe. This current density is comparable to the critical current density of $j = 12.75 \times 10^{10} \text{ A/m}^2$ in our experiments, which is required to generate the skyrmions through Joule heating. Our j - v curve was measured at current densities below $j = 10 \times 10^{10} \text{ A/m}^2$ under the pulse duration of 5 ns. At this level, the maximum local temperature reaches $\sim 180 \text{ K}$, far below the Curie temperature. At $j = 7 \times 10^{10} \text{ A/m}^2$, the maximum temperature increase is only $\sim 40 \text{ K}$. Given the linear relationship between current density and velocity below $j = 10 \times 10^{10} \text{ A/m}^2$, we believe that the local Joule heating effects are not significant within the reasonable range of current densities in **Fig. 2**.

Additionally, in our experiments at higher current densities, the signature of local heating becomes apparent. As shown in **Video 3** at a current density of $\sim 11 \times 10^{10} \text{ A/m}^2$ and pulse duration of 5 ns, the skyrmion is observed to be annihilated during its motion. At an even higher current density of $\sim 12 \times 10^{10} \text{ A/m}^2$, the skyrmion is annihilated as soon as the current is applied. These observations are consistent with our simulations of local heating effects.

Fig. R12 | Simulated local heating effects in a 100-nm-wide FeGe nanostripe using COMSOL Multiphysics software. a, Schematic of the device geometry for simulation. **b,** Temperature profiles are plotted over time. The pulse duration is 5 ns. The vertical dotted line indicates the time at 5 ns, after which the current is turned off.

Revision:

The simulated results of local heating effects and related discussions have been incorporated into the revised manuscript (**Methods: Estimate of local heating effects**) and supplementary information.

“**Estimate of local heating effects.** The local heating effects were concerned during our experiments. We utilized COMSOL simulations to evaluate these effects. As shown in **Supplementary Fig. 16**, with a current density of $j = 13 \times 10^{10} \text{ A/m}^2$ and a pulse duration of 5 ns, the local temperature reaches 260 K, approaching the Curie temperature of FeGe. This current density is comparable to the critical current density of $j = 12.75 \times 10^{10} \text{ A/m}^2$ in our experiments, which is required to generate the skyrmions through Joule heating in **Supplementary Fig. 4**. Our j - v curve was measured at current densities below $j = 10 \times 10^{10} \text{ A/m}^2$ at the pulse duration of 5 ns. At this level, the maximum local temperature reaches ~ 180 K, far below the Curie temperature. At $j = 7 \times 10^{10} \text{ A/m}^2$, the maximum temperature increase is only ~ 40 K. Given the linear relationship between current density and velocity below $j = 10 \times 10^{10} \text{ A/m}^2$, we believe that the local Joule heating effects are not significant within the reasonable range of current densities in **Fig. 2**. Additionally, at higher current densities, the signature of local heating became apparent. As shown in **Video 3** at a current density of $\sim 11 \times 10^{10} \text{ A/m}^2$ and pulse duration of 5 ns, the skyrmion was observed to be annihilated during its motion. At an even higher current density of $\sim 12 \times 10^{10} \text{ A/m}^2$, the skyrmion was annihilated as soon as the current was applied.”

Comment 3: What is the explanation for the skyrmion being pinned more easily for smaller current pulses (Fig. S6)? Is there a fundamental lower limit on pulse-length required to drive the skyrmion motion?

Response:

We thank the reviewer for his/her comments. The phenomenon of skyrmion being more easily pinned for smaller current pulses can indeed be understood through the picture of pinning effects. The pinning centers can be considered as effective potential wells with a characteristic size. Moving the skyrmion's center across this pinning potential takes time, which leads to easier pinning when shorter current pulses are used.

Depinning of skyrmions is influenced by both the current density and the pulse width, as addressed in a previous study [PRB 109, 104405 (2024)]. The critical current density required to initiate skyrmion movement decreases exponentially with increasing pulse width, as demonstrated experimentally in **Fig. 2**. This relationship indicates that there is a fundamental lower limit on the pulse length necessary to drive skyrmion motion effectively. However, the fundamental lower limit on pulse length is contingent on the density and strength of local pinning centers in real material, which is determined by the quality of the crystals.

Revision:

These following statements have been involved in the revised manuscript (**Section: Current-driven motion of a single skyrmion in a 100-nm-wide FeGe nanostripe**),

“The critical current density j_c to initiate skyrmion movement decreases exponentially with increasing pulse duration (**Fig. 2d**). At $\tau = 2$ ns, j_c is $\sim 7.89 \times 10^{10} \text{ A} \cdot \text{m}^{-2}$. It is reduced to $\sim 3.67 \times 10^{10} \text{ A} \cdot \text{m}^{-2}$ at $\tau = 8$ ns. Below 2 ns, the skyrmion is pinned more easily (**Supplementary Fig. 6** and Supplementary Video 5). Depinning of skyrmions is influenced by both the current density and the pulse width. The phenomenon of skyrmion being more easily pinned for smaller current pulses can indeed be understood through the picture of pinning effects. The pinning centers can be considered as effective potential wells with a characteristic size. Moving the skyrmion's center across this pinning potential takes time, which leads to easier pinning when shorter current pulses are used. The fundamental lower limit on pulse length is contingent on the density and strength of local pinning centers in real material.”

Comment 4: The ratio of nonadiabatic spin-transfer torque and Gilbert damping parameter is noted to be considerably larger than the theoretical prediction. Can the authors explain the significance of this and why the experimental determination is so significantly larger than previous theory?

Response:

We thank the reviewer for his/her insightful comments. This is indeed a general question that we also wish to address, as the ratio of nonadiabatic spin-transfer torque and Gilbert damping parameter critically

influences the dynamics in 1D scenarios for both skyrmions and domain walls. However, a well-established theory to comprehensively describe nonadiabatic spin-transfer torque remains absent. There are some existing measurements of beta or beta/alpha for materials like permalloy documented in the literatures, yet a clear physical model is still not well established. In our previous preprint [10.48550/arXiv.2212.08991], we attempted to correlate the nonadiabatic spin-transfer torque with factors such as the density of states (DOS), saturation magnetization (M_s), spin polarization (P) and the spin splitting energy (E), acknowledging that the theoretical framework for these correlations is still under debate. We aim to provide further explanations and developments in our subsequent works.

Comment 5: Does the current-velocity relation change for the cases of skyrmion chains and how does this change with an increasing number of skyrmions?

Response:

We thank the reviewer for his/her comments. The skyrmion chains almost exhibit the same current-velocity relation to that for single skyrmion in the nanostripe. As shown in **Fig. R13** below, we have measured the velocities of skyrmion chains with various numbers of skyrmions ($N = 2, 4$ and 7) in the experiments, and compared them with that of single skyrmions ($N = 1$). The j - v curves are consistent regardless of the skyrmion number, indicating that both single skyrmion and skyrmion chains follow similar current-driven dynamics.

Furthermore, we have performed micromagnetic simulations to investigate whether the skyrmion velocity is influenced by the number of skyrmions in chains. As shown in **Fig. R14** below, the velocities of skyrmion chains are almost independent of the number of skyrmions, with variation primarily due to pinning effects and the limited number of current pulses applied in the simulations. The results are consistent with our experimental observations.

Fig. R13 | Skyrmion velocities plotted as a function of current density. The skyrmion chains with various skyrmion number are plotted together for comparison. The pulse duration is 5 ns.

Fig. R14 | Simulated velocities of skyrmion chains with a various number of skyrmion. The current densities are $5 \times 10^{10} \text{ A/m}^2$ (orange) and $10 \times 10^{10} \text{ A/m}^2$ (green), respectively. The pulse duration is 5 ns. The velocities are averaged over five current pulses in the simulations. The variations are primarily due to pinning effects and the limited number of current pulses applied in the simulations.

Revision:

The following discussions and new results (**Figs. R13 and R14**) have been incorporated in the revised manuscript (**Section: Single-chain skyrmions and their current-driven motion**) as follows,

“Skyrmion chains, featuring various numbers of skyrmions, could be steadily driven by currents (**Fig. 4b-c, Supplementary Fig. 11, Supplementary Videos 10 and 11**), exhibiting the same one-dimensional motion as that observed for a single skyrmion. The measured j - v curves are almost consistent across different skyrmion numbers (**Supplementary Fig. 12**), indicating that both single skyrmion and skyrmion chains follow similar current-driven dynamics. This behavior is further reproduced by micromagnetic simulations. The skyrmion chains with different numbers of skyrmions, where the inter-skyrmion interaction is attractive at this distance, show that the current-driven motion remains translational along the track, following the similar inertia motion (**Supplementary Fig. 13**). Additionally, the corresponding velocities are almost independent of the number of skyrmions (**Supplementary Fig. 14**), with variations primarily due to pinning effects and the limited number of current pulses applied in the simulations. These observations demonstrate that the skyrmion-skyrmion interaction does not significantly alter the dynamics of the skyrmion chains.”

Comment 6: How does the skyrmion inertia change for a collective skyrmion chain? Can the inertia still considered for the single skyrmion or does the object need to be considered collectively?

Response:

We thank the reviewer for his/her comments. The skyrmion inertia picture does work for the skyrmion chain case. Previous study found that the interaction between skyrmions shows a Lennard-Jones-type behavior [PRL 120, 197203 (2018)], i.e., below (above) a critical distance, the skyrmions attract (repel) each other. Although the skyrmion chains may possess other degrees of freedom in a 2D system, in a racetrack-type device like the one studied in our work, due to the confinement of the track, the dynamics of multiple skyrmions reduces to a translational motion along the track. We have performed further micromagnetic simulations to check the inertia motion in skyrmion chains with various skyrmion number. As shown in **Fig. R15**, for different numbers of skyrmions, the current-driven skyrmion motion is a translation along the track, showing the similar inertia motion after the current pulse is turned off. The variations are primarily caused by pinning effects. Furthermore, the velocities are almost independent of the number of skyrmions as shown in **Fig. R14** above. Therefore, the inertia picture can also apply to the case of a skyrmion chain.

Fig. R15 | Skymion displacement ΔX plotted as a function of time for skymion chains with various skymion number (N). The pulse duration is 5 ns and the current density is $j = 10 \times 10^{10}$ A/m². The vertical dashed line indicates a time of 5 ns. The variations in displacements are caused by the randomly distributed pinning sites modeled in the nanostripe.

Revision:

The following discussions and new results (**Figs. R15**) have been incorporated in the revised manuscript (**Section: Single-chain skyrmions and their current-driven motion**) as follows,

“Skyrmion chains, featuring various numbers of skyrmions, could be steadily driven by currents (Fig. 4b-c, **Supplementary Fig. 11**, Supplementary Videos 10 and 11), exhibiting the same one-dimensional motion as that observed for a single skyrmion. The measured j - v curves are almost consistent across different skyrmion numbers (**Supplementary Fig. 12**), indicating that both single skyrmion and skyrmion chains follow similar current-driven dynamics. This behavior is further reproduced by micromagnetic simulations. The skyrmion chains with different numbers of skyrmions, where the inter-skyrmion interaction is attractive at this distance, show that the current-driven motion remains translational along the track, following the similar inertia motion (**Supplementary Fig. 13**). Additionally, the corresponding velocities are almost independent of the number of skyrmions (**Supplementary Fig. 14**), with variations primarily due to pinning effects and the limited number of current pulses applied in the simulations. These observations demonstrate that the skyrmion-skyrmion interaction does not significantly alter the dynamics of the skyrmion chains.”

Comment 7: For the collective motion of a skyrmion chain (Fig. S9), the skyrmion-skyrmion interactions leads to non-synchronous motion which would likely lead to difficulties in extending this to racetrack-type device schemes. Can the authors determine a minimum skyrmion-skyrmion spacing before this non-synchronous motion occurs and therefore state a maximum information-density possible within this nanotrack regime?

Response:

We thank the reviewer for his/her insightful comments. As the reviewer noted, skyrmion-skyrmion interactions can lead to non-synchronous motion, necessitating a minimum skyrmion-skyrmion spacing to avoid this phenomenon. However, this process is also influenced by pinning effects in real materials, such as the density and strength of pinning sites, which can likewise induce non-synchronous motion. Consequently, it is not easy to estimate the maximum information-density when considering these pinning effects. Nevertheless, excluding the influence of pinning effects, it is possible to give a rough estimate of the areal density for the system under investigation. Based on the model proposed in a previous work [PRL 120, 197203 (2018)], skyrmion-skyrmion interaction exhibits a Lennard-Jones-type behavior, i.e., below (above) a critical distance, the skyrmions attract (repel) each other. The safe skyrmion-skyrmion spacing would likely need to exceed 200 nm in FeGe. This roughly corresponds to a maximum information-density of ~ 4 Gb/inch² in a 100-nm-wide nanotrack.

The density can be further increased from several different perspectives. First, the minimum spacing is closely related to the size of skyrmion. For materials with skyrmion sizes around several nanometers, the density can be increased significantly. The position of skyrmion can also be controlled by external means such as artificial pinning sites (e.g., patterning notches along the edges). Therefore, the minimum skyrmion-skyrmion spacing to avoid the non-synchronous motion may be further reduced.

Revision:

We have involved the following statements in the revised manuscript (**Section: Single-chain skyrmions and their current-driven motion**) and supplementary information,

“As the spacing decreases, prominent skyrmion-skyrmion interactions [36] comes into play. This is expected to result in varying displacements for each skyrmion in the presence of an electrical current, leading to the non-synchronous motion (**Supplementary Fig. 10**). Therefore, it necessitates a minimum skyrmion-skyrmion spacing to avoid this phenomenon. However, this process is also influenced by pinning effects in real materials, such as the density and strength of pinning sites, which can likewise induce non-synchronous motion (see **Supplementary Note III** for the discussion of information-density in the 100-nm-wide nanostripe)”.

Supplementary Note III – Estimate of information-density in the 100-nm-wide nanotrack

It is possible to give a rough estimate of the areal density disregarding the influence of pinning effects. Based on the skyrmion-skyrmion interactions described before [36], the safe skyrmion-skyrmion spacing would likely need to exceed 200 nm in FeGe. This roughly corresponds to a maximum information-density of approximately 4Gb/inch² in a 100-nm-wide nanotrack. The density can be further increased by selecting materials with smaller skyrmion sizes. The minimum spacing is closely related to the size of skyrmion. For materials with skyrmion sizes around several nanometers, the density can be increased significantly. The position of skyrmion can also be controlled by external means such as artificial pinning sites (e.g., patterning notches along the edges). Therefore, the minimum skyrmion-skyrmion spacing to avoid the non-synchronous motion may be further reduced.

Comment 8: In 496080_0_video_73316_s9wnwz.mp4 and 496080_0_video_73317_s9wnx0.mp4, at the 14th and 2nd pulses respectively, it looks like the skyrmion splits into two. Can the authors explain this?

Response:

We thank the reviewer for his/her scurrility and comments. Indeed, the apparent splitting of skyrmions observed in the videos was caused by the low temporal resolution in our experiments. The videos were recorded with a frame rate of 0.2 seconds, and the frequency of the current pulse is 1 Hz. Consequently, five images were included to record the motion at each current pulse. However, due to a lack of synchronization between the start of the recording and the applied current pulse, overlapping of the skyrmion between two images occurred.

To avoid this confusion and provide a clearer representation, we have selected another frame from the available five frames to construct the videos. This adjustment helps eliminate the misleading appearance of skyrmion splitting.

Revision:

The issues with the two videos have been fixed in the revised manuscript.

REVIEWER COMMENTS

Reviewer #1 (Remarks to the Author):

the authors have properly answered all the question raised by reviewers. The paper can be accepted for publication.

Reviewer #2 (Remarks to the Author):

The reviewer is satisfied with the revisions and believe that this work should be published ASAP.

Reviewer #3 (Remarks to the Author):

The authors have taken into account the referees' comments and made appropriate adjustments to the manuscript. One last question regarding the consistency of displacement per pulse for the pulsewidth series in Fig. S15. It is clear that the displacement per pulse varies slightly along the pulse series, likely due to random pinning sites. However, by calculating the mean displacement per pulse and standard deviation for each of the tested current densities, can the authors potentially quote an optimal current density for consistent displacement per pulse? This may then work as an estimated optimal current density for any racetrack type device which requires consistent skyrmion placement.

RE: Manuscript Number: NCOMMS-24-17355A

“Steady motion of 80-nm-size skyrmions in a 100-nm-wide track” by Dongsheng Song et al.

Reviewer #1 (Remarks to the Author):

The authors have properly answered all the question raised by reviewers. The paper can be accepted for publication.

Response:

We appreciate the reviewer's support in publishing our work.

Reviewer #2 (Remarks to the Author):

The reviewer is satisfied with the revisions and believe that this work should be published ASAP.

Response:

We appreciate the reviewer's support in publishing our work.

Reviewer #3 (Remarks to the Author):

The authors have taken into account the referees' comments and made appropriate adjustments to the manuscript. One last question regarding the consistency of displacement per pulse for the pulse width series in Fig. S15. It is clear that the displacement per pulse varies slightly along the pulse series, likely due to random pinning sites. However, by calculating the mean displacement per pulse and standard deviation for each of the tested current densities, can the authors potentially quote an optimal current density for consistent displacement per pulse? This may then work as an estimated optimal current density for any racetrack type device which requires consistent skyrmion placement.

Response:

We appreciate the reviewer's support in publishing our work and thank the reviewer for his/her insightful comment. Regarding the optimal current density for consistent skyrmion motion, we have plotted the ratio of skyrmion velocities (v) and their standard deviations (Δv) in **Fig. R1** under various current densities and pulse durations (τ). The optimal current density can be identified by a lower ratio, marked by the green box with a ratio below 5%. Specifically, the optimal current density is approximately between 10×10^{10} A/m² and 12×10^{10} A/m² at $\tau = 2$ ns, between 8×10^{10} A/m² and 11×10^{10} A/m² at $\tau = 3$ ns, between 7×10^{10} A/m² and 9×10^{10} A/m² at $\tau = 4$ ns, between 6.5×10^{10} A/m² and 9×10^{10} A/m² at $\tau = 5$ ns, between 5×10^{10} A/m² and 8.5×10^{10} A/m² at $\tau = 6$ ns, and between 4.5×10^{10} A/m² and 7.5×10^{10} A/m² at $\tau = 8$ ns.

Revision:

Fig. R1 have been involved in the revised supplementary information as **Supplementary Fig. 16**.

In the **Methods** part, **Calculation of skyrmion velocities**, we have involved the following statements,

“By calculating the mean velocities and standard deviations under various current densities and pulse durations (τ), we can quote the range of the optimal current density for consistent displacement. The ratio of skyrmion velocities (v) and their standard deviations (Δv) is plotted in **Supplementary Fig. 16** under various current densities and pulse durations. The optimal current density can be identified by a lower ratio, marked by the green box with a ratio below 5%. Specifically, the optimal current density is approximately between 10×10^{10} A/m² and 12×10^{10} A/m² at $\tau = 2$ ns, between 8×10^{10} A/m² and 11×10^{10} A/m² at $\tau = 3$ ns, between 7×10^{10} A/m² and 9×10^{10} A/m² at $\tau = 4$ ns, between 6.5×10^{10} A/m² and 9×10^{10} A/m² at $\tau = 5$ ns, between 5×10^{10} A/m² and 8.5×10^{10} A/m² at $\tau = 6$ ns, and between 4.5×10^{10} A/m² and 7.5×10^{10} A/m² at $\tau = 8$ ns.”

Fig. R1 | The ratio of skyrmion velocities (v) and their standard deviations (Δv) under various current densities and pulse durations. Values below 5% are highlighted in the green box for reference.

REVIEWERS' COMMENTS

Reviewer #3 (Remarks to the Author):

I am satisfied with all revisions and believe the paper should be accepted for publication.